

# Projecting changes in rainfall-induced landslide susceptibility across China under climate change

Jinqi Wang [1, 2], Hao Fang [2, 3, 4], Kai Liu [1, 2*], Yi Yue [2], Ming Wang [1, 2], Bohao Li [1, 2], Xiaoyi Miao [1, 2]

[1]Joint International Research Laboratory of Catastrophe Simulation and Systemic Risk Governance, Beijing Normal University, Zhuhai 519087, China
[2] School of National Safety and Emergency Management, Beijing Normal University, Beijing 100875, China
[3] Zhejiang Institute of Geosciences, Hangzhou 310007, China
[4] Qiantang Institute of Geosciences, Hangzhou 310007, China

*Correspondence to*: Kai Liu (liukai@bnu.edu.cn)

**Abstract** Landslides pose a significant threat to human lives and property. Evaluating dynamic changes in landslide susceptibility under climate change can provide decision-making support for future disaster prevention. Using historical landslide inventories (2008-2023) and a random forest algorithm, this study develops an annual-scale landslide susceptibility model to assess spatiotemporal patterns of landslide susceptibility across China under different simulated scenarios. The results show that model achieves excellent performance (AUC = 0.97), with annual precipitation being the most influential factor (26% contribution). Compared to the baseline (1950–2014), China's landslide susceptibility is projected to increase significantly under future climate conditions. By the late 21st century (2076–2100), the national mean annual precipitation is expected to rise by 59–111 mm, corresponding to a 4.3–10.6% expansion in median to very high susceptibility zones across SSP scenarios. Spatially, the most significant susceptibility increases are anticipated in the Northwest Loess Plateau region (Loess) near the Taihang Mountains and the northern part of the Southwest Karst Mountain region (SW), where SSP5-8.5 amplify risks toward the century's end. These findings underscore the necessity of proactive risk management in these identified hotspots to mitigate escalating landslide threats.

## 1 Introduction

Rainfall-induced landslides rank among the world's most destructive geological disasters (Gutiérrez et al., 2015), having claimed over 48,000 documented lives globally between 2004 and 2016 (Emberson et al., 2020, 2021; Froude and Petley, 2018). China faces particularly severe impacts, with landslide-related fatalities averaging 600 annual deaths since 2000—accounting for nearly 25% of the country's natural disaster mortality (Lin et al., 2022; Lin and Wang, 2018). Climate change is expected to exacerbate this threat through increased frequency and intensity of extreme precipitation events (Gariano et al., 2017; Kirschbaum et al., 2020), potentially elevating regional landslide susceptibility and associated risks. A comprehensive understanding of future susceptibility patterns is therefore critical for developing effective, climate-resilient landslide mitigation strategies.



Landslide occurrence results from the interplay between static predisposing factors and dynamic triggering conditions. Recent advances in susceptibility modelling have increasingly incorporated time-dependent variables to better capture spatiotemporal patterns of landslide risk. Currently, landslide susceptibility mapping (LSM) methodologies fall into four main categories: physical models, expert-driven models, statistical models, and machine learning models (Chang et al., 2019; Li et al., 2019).

Among these, machine learning algorithms have demonstrated particular effectiveness in landslide susceptibility modelling due to their capability to capture both linear and nonlinear relationships between causative factors and landslide occurrence (Merghadi et al., 2020). Commonly employed machine learning approaches include random forest (RF), support vector machines (SVM), Extreme gradient boosting (XGBoost), convolutional neural networks (CNN), logistic regression (LR), k-nearest neighbours (KNN), and long short-term memory networks (LSTM) (Avand et al., 2019; Hakim et al., 2022; Huang et

al., 2023; Jin et al., 2022; Zhang et al., 2023). RF algorithm has gained particular attention in this field due to its distinctive advantages. First, its parallel computing architecture enables efficient processing of large datasets, while the ensemble approach of multiple decision trees significantly enhances both training efficiency and prediction accuracy. Second, RF incorporates a built-in feature importance evaluation module that assesses variable significance through permutation-based comparison with prediction outcomes (Zhao et al., 2018). Extensive studies have confirmed RF's superior performance in

landslide susceptibility assessments (Chen et al., 2017; Dou et al., 2019; Hong et al., 2019; Reichenbach et al., 2018).

Recent studies assessing landslide susceptibility under future climate change scenarios typically rely on historical landslide inventories with temporal information, using the spatial distribution of past events as training and validation data (Duan et al., 2025; Semnani et al., 2025). These inventories are integrated with multi-source conditioning variables—such as meteorological, topographic, and land use factors—for quantitative modelling (Merghadi et al., 2020). On this basis, future scenario outputs

from Coupled Model Intercomparison Project Phase 6 (CMIP6) climate models (such as precipitation and temperature) are often input into susceptibility models to predict the spatial distribution and temporal trends of future landslide susceptibility. However, this modelling framework faces several widely recognized limitations. On the one hand, the incompleteness and spatial-temporal biases of historical inventories—due to limited event reporting and archival standards—can undermine the representativeness of training samples and inject substantial uncertainty into model predictions (Lin et al., 2021). This issue

may be more pronounced at the national scale (Du et al., 2020). On the other hand, the future climate projections from CMIP6 feature relatively low spatial resolution, and inter-model differences persist among different simulation outputs (Dobler et al., 2024; John et al., 2022). This can further increase the uncertainty of future landslide susceptibility predictions (Reichenbach et al., 2018).

To mitigate these uncertainties, this study developed an annual landslide susceptibility model based on 2008-2023 geological

disaster records from the Ministry of Natural Resources of China. The model employs a RF algorithm with feature engineering, combined with statistically downscaled climate projections from NASA Earth Exchange Global Daily Downscaled Projections (NEX-GDDP-CMIP6) to quantify the future changes in landslide susceptibility across China under climate change scenarios. The results can provide critical insights for climate-adaptive land management and disaster risk reduction strategies in vulnerable regions.



## 2 Materials

### 2.1 Landslide inventory

The reliability of the analysis largely depends on the accuracy of historical landslide disaster records. In this study, geological disaster records from 2008 to 2023, provided by the Ministry of Natural Resources of the People's Republic of China, are used as the foundational data for model training and future trend projections. These records include information on three types of geological disasters—landslides, debris flows, and collapses—including the date of occurrence, location (longitude and latitude). In this study, these three types of disasters are collectively referred to as "landslides". Fig. 1 presents the spatial distribution of landslide events in the dataset.

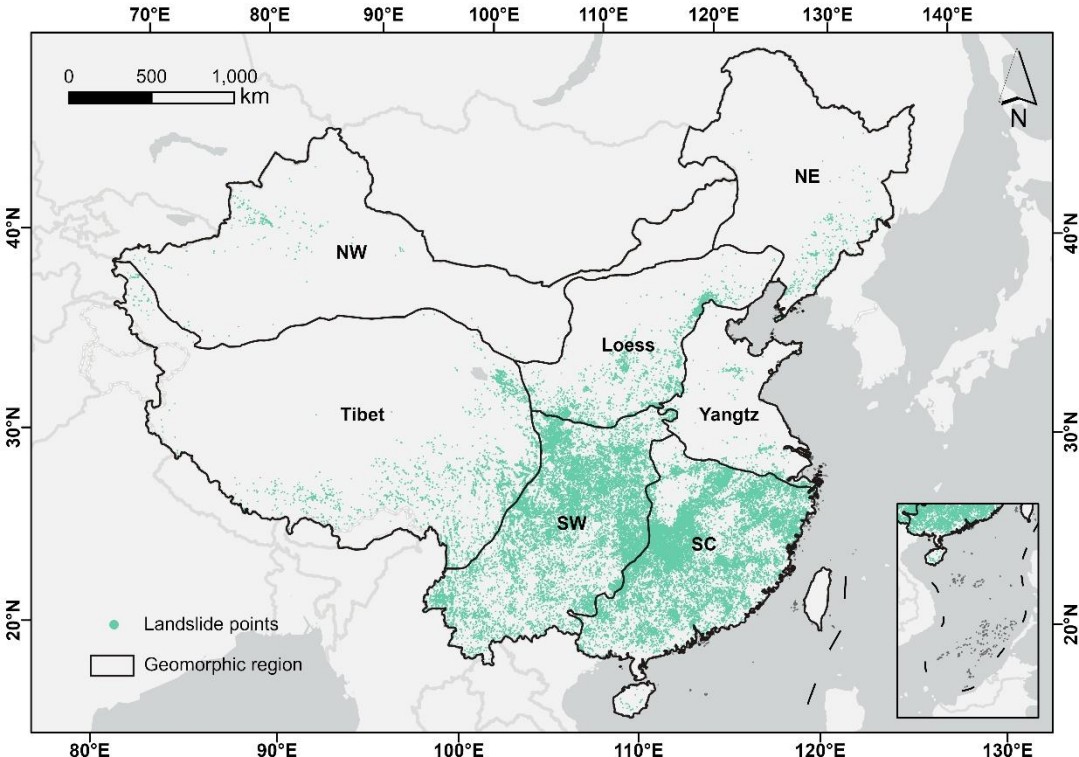

**Figure. 1:** Spatial distribution of rainfall-induced geological disaster events and major geological environmental regions in China. Geological environmental regions boundaries were obtained from https://geocloud.cgs.gov.cn/. NE: Northeastern wetland ecology region; Yangtz: Huang-Huaihai-Yangtze River Delta plain region; SC: South China bedrock hills region; Loess: Northwest Loess Plateau region; SW: Southwest karst mountain region; NW: Northwest arid desert region; Tibet: permafrost region of the Qinghai-Tibet Plateau.



## 2.2 Landslide influencing factors

In this study, we selected 14 factors that may influence landslide susceptibility (Table 1). All variables were resampled to a resolution of 1 km × 1 km. For continuous variables such as elevation and distance to river, we used bilinear interpolation, while for discrete variables including lithology and land use type, we employed mode sampling.


**Table 1**. Datasets and their corresponding landslide influencing factors. "-" indicates no time-series information.

| Data Type | Dataset | Resolution | Time Period | Influencing Factor |
|---|---|---|---|---|
| Geological Features | GLiM | 1:3750,000 | - | Lithology |
| | GEM GAF-DB | Vectors | - | Distance to fault |
| | Global Seismic Hazard Map | 0.04° | - | 475-year return period PGA |
| Geomorphometric Features | SRTM Digital Elevation Data Version4 | 90 m | - | Elevation<br>Slope<br>TWI<br>Curvature<br>Plan curvature<br>Profile curvature |
| Hydrological Features | 1-km Monthly Precipitation Dataset for China | 1 km | 1950-2023 | Annual total precipitation |
| | NEX-GDDP-CMIP6 | 0.25° | 1950-2100 | |
| | Five-level River Dataset of China | 1:1,000,000 | - | Distance to river |
| Environmental Features | China's Land-Use/Cover Datasets (CLCD) | 30 m | 2008-2023 | Landcover |
| | MOD13Q1.006 Terra Vegetation Indices 16-DayGlobal250m | 250 m | 2008-2023 | NDVI |
| | Open Street Map Global Primary Roads | Vectors | - | Distance to road |

### 2.2.1 Geological factors

Fault structures and lithological characteristics may affect slope stability by weakening the structural integrity of rock masses. Under the combined influence of earthquakes and heavy rainfall, geologically vulnerable areas are more likely to experience

slope failure. This research combines three geological elements. First, the distance to fault was calculated. This calculation used the Euclidean Distance tool in ArcGIS Pro. The data were sourced from the Global Active Faults Database (GAF-DB),





which contains spatial information for 13,500 faults (Styron and Pagani, 2020). Second, lithology was represented using the Global Lithological Map (GLiM), compiled by the International Union of Geological Sciences (IUGS). This dataset has a scale of 1:3,750,000. It describes the engineering geological properties of surface rock layers (Hartmann and Moosdorf, 2012).

Third, the 475-year return period peak ground acceleration ((475-year RP PGA)) was used. It reflects the long-term future cumulative damage from seismic energy on slope rock masses. The 475-year RP PGA spatial distribution data came from Johnson et al., (2023).

### 2.2.2 Geomorphometric factors

Geomorphometric factors are the main factors controlling the spatial distribution of surface runoff and soil moisture movement.
They are also key triggers for landslide disasters. This study selects five geomorphological factors: elevation, slope, curvature, plan curvature, and profile curvature. The elevation data are sourced from the fourth edition of the Shuttle Radar Topography Mission (SRTM V4) digital elevation database provided by NASA (Jarvis et al., 2008), and the other four topographic factors were calculated based on ArcGIS Pro.

### 2.2.3 Hydrological factors

We selected three hydrological factors for this study: annual total precipitation, distance to river, and the topographic wetness index (TWI). Precipitation data were obtained from the "China 1 km Resolution Monthly Precipitation Dataset (1901–2023)" provided by the Qinghai-Tibet Plateau Science Data Centre (Ding and Peng, 2020; Peng et al., 2017, 2018, 2019). TWI quantifies surface moisture conditions by integrating upslope contributing area and local slope. High TWI values typically indicate zones of soil moisture accumulation, which are more susceptible to landslides. In this study, TWI was calculated based
on SRTM V4 using the hydrological analysis tools in ArcGIS Pro. The river system affects slope stability through two primary mechanisms. First, riverbank erosion can compromise slope structural integrity. Second, the distance to river factor indirectly modulates slope instability by influencing surface runoff energy and erosion intensity. We employed the five-level river dataset of China at a 1:1,000,000 scale, which is available from. We then generated the distance-to-river map using the Euclidean Distance tool in ArcGIS Pro.

### 2.2.4 Environmental factors

Land cover and Normalized Difference Vegetation Index (NDVI) influence the occurrence of landslides by affecting surface runoff and infiltration. For example, areas with extensive vegetation cover have more permeable surfaces, while areas with urban land have large impermeable surfaces and lower infiltration rates. Land cover data is sourced from China's Land-Use/Cover Datasets (Yang and Huang, 2023). The NDVI was calculated by averaging all available MOD13Q1 V6 NDVI
images from 2008 to 2023 in Google Earth Engine (Didan, 2021). Human activities may indirectly affect landslides by altering topography and land cover. For instance, road construction often involves cutting through slopes, weakening slope stability, and damaging soil and rock structures, which increases landslide susceptibility. In this study, road network data for China was





obtained from OpenStreetMap (www.openstreetmap.org). The distance to road was then calculated using the Euclidean Distance tool in ArcGIS Pro.

### 2.2.5 Simulation data

High-resolution climate projections from the NEX-GDDP-CMIP6 dataset (0.25° spatial resolution) were used to drive the simulation of future landslide susceptibility in China. (Thrasher et al., 2022). This study comprehensively selects three typical shared socioeconomic pathway (SSP) scenarios: SSP1-2.6 (sustainable development path), SSP2-4.5 (middle-of-the-road path), and SSP5-8.5 (high carbon emission path). These scenarios cover simulation output data from 31 General Circulation Models (GCMs). The study compares the spatial and temporal variation characteristics of mean annual precipitation during the historical baseline period (1950-2014) and future periods: near-term future (2026-2050), mid-term future (2051-2075), and long-term future (2076-2100).

## 3 Methodology

### 3.1 Framework for this work

The assessment of rainfall-triggered landslide susceptibility in China under climate change is structured into six key steps (Fig. 2): (1) extracting landslide points from the inventory and generating non-landslide points by random sampling outside buffer zones around landslide locations; (2) applying feature selection techniques to identify the conditioning factors from landslide conditioning variables; (3) constructing a feature matrix by extracting static and dynamic conditioning factors at each sample point, and dividing the dataset into training (70%) and testing (30%) sets; (4) training and optimizing an annual landslide susceptibility model using the RF algorithm; (5) LSM for both historical and future periods by applying downscaled climate projections under different climate change scenarios; (6) analysing the spatiotemporal changes in landslide susceptibility across China under various climate change scenarios.



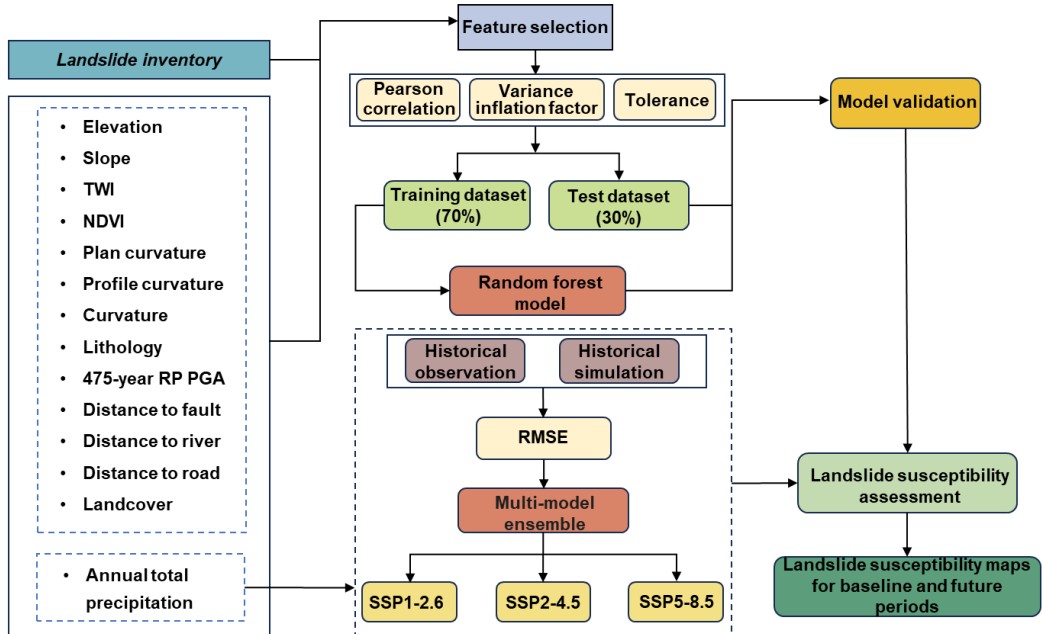

**Figure. 2:** Methodological framework of this study

## 3.2 Extracting disaster samples and generation of non-disaster samples

All landslide records were spatialized in ArcGIS Pro, resulting in 93,935 landslide points. For each landslide point, a 3 km buffer was established. Twice as many non-landslide points were randomly generated within the land area of China and outside all buffer zones. To ensure temporal consistency, each non-landslide point was randomly assigned a year between 2008 and 2023, matching the temporal distribution of the landslide points.

## 3.3 Feature selection

Feature selection is an essential preprocessing step for high-dimensional data analysis, visualization, and modelling. In this study, two methods were employed for feature selection to enhance computational efficiency while preserving the accuracy of the model: Pearson correlation coefficient and tolerance (TOL) and variance inflation factor (VIF).

### 3.3.1. Pearson correlation coefficient

To quantify the linear relationship between features, the Pearson correlation coefficient was computed. A correlation was considered strong and statistically significant if the absolute value of the correlation coefficient (|r|) exceeded 0.7 and the associated p-value was less than 0.05. When such highly correlated features were identified, we addressed potential





multicollinearity by removing one of the redundant features. This approach provides a quantitative means of identifying and mitigating multicollinearity among input features (Li et al., 2022).

$$r = \frac{1}{n-1}\sum_{i=1}^{n}\left(\frac{X_i-\bar{X}}{\sigma_x}\right)\left(\frac{Y_i-\bar{Y}}{\sigma_y}\right),\tag{1}$$

Here, $r$ is the Pearson correlation coefficient; $n$ is the sample size; $X$ and $Y$ are the variables; $\bar{X}$ and $\bar{Y}$ are the sample means of $X$ and $Y$, respectively; $\sigma_x$ and $\sigma_y$ are the sample standard deviations.

**3.3.2. Tolerance and variance inflation factor**

A strong linear relationship between two influencing factors may reduce a model's predictive accuracy. To diagnose potential multicollinearity, TOL and VIF are widely used. Multicollinearity is indicated when TOL < 0.1 and VIF > 10. In such cases, variables exhibiting high collinearity should be removed (He et al., 2021; O'brien, 2007; Roy and Saha, 2019). The specific formulas are as follows.

$$TOL = 1 - R_i^2\,(i = 1,2,\cdots m)\,,\tag{2}$$

$$VIF = \frac{1}{1-R_i^2}\,,\tag{3}$$

Here, $R_i^2$ represents the coefficient of determination obtained by regressing the $i$th independent variable on the remaining $m - 1$ independent variables.

**3.4 Generation of a feature matrix**

Relevant features were extracted for each data point. Dynamic variables were obtained by constructing a time-series three-dimensional matrix (time × location × variable) in Python. Static variables, such as elevation and slope, were extracted using the Extract Multi Values to Points tool in ArcGIS Pro. All extracted variables were then integrated into a single feature matrix for model training.

**3.5 RF algorithm**

RF is an ensemble learning algorithm that improves predictive performance by building and aggregating multiple decision trees (Breiman, 2001). The algorithm employs bootstrap aggregation, whereby each tree is trained on a randomly drawn bootstrap sample from the original dataset, and at each node, a random subset of features is selected for splitting. This randomization process promotes model diversity and robustness, enhancing accuracy and reducing sensitivity to noise and outliers (Stumpf and Kerle, 2011). By averaging the results of multiple trees, RF effectively reduces variance and the risk of

overfitting, making it more robust than individual decision trees. In addition, RF can efficiently handle high-dimensional datasets and categorical variables with minimal preprocessing or encoding. Importantly, RF provides feature importance



measures, which facilitate the identification of key variables influencing model predictions (Bureau et al., 2003). These strengths make RF particularly suitable for tasks such as landslide susceptibility mapping, where high accuracy, robustness, and interpretability are essential.

## 3.6 Model training and performance evaluation

Accurate evaluation of the RF model is essential for predicting future changes in landslide susceptibility. In this study, we employed several widely used evaluation metrics, including accuracy, precision, recall, F1-score, the receiver operating characteristic (ROC) curve, and the area under the curve (AUC). These metrics are derived from the confusion matrix (M and M.N, 2015), with values closer to 1 indicating better model performance. Specifically, the ROC curve visualizes the trade-off between the false positive rate (FPR) on the x-axis and the true positive rate (TPR) on the y-axis.

For model development and validation, we randomly split the dataset into training (70%) and testing (30%) subsets. RF model was trained on the training subset, and all performance metrics were evaluated on the independent testing subset to avoid overfitting. Model hyperparameters were optimized and set to n_estimators = 886, max_depth = 20, max_features = "sqrt", and min_samples_split = 6, using the Gini index as the splitting criterion.

$$Accuracy = \frac{TP+TN}{TP+TN+FP+FN}, \tag{4}$$

$$Precision = \frac{TP}{TP+FP}, \tag{5}$$

$$Recall = TPR = \frac{TP}{TP+FN}, \tag{6}$$

$$F1 - score = \frac{2TP}{2TP+FP+FN}, \tag{7}$$

$$FPR = \frac{FP}{FP+TN}, \tag{8}$$

Here, TP (true positive) and TN (true negative) represent the number of landslide and non-landslide samples that are correctly classified, respectively. FP (false positive) refers to the number of non-landslide samples that are incorrectly classified as landslides, while FN (false negative) refers to the number of landslide samples that are incorrectly classified as non-landslides.

## 3.7 Assessment of future landslide susceptibility

We selected 31 GCMs from the NEX-GDDP-CMIP6 dataset. Each of these models includes the SSP1-2.6, SSP2-4.5, and SSP5-8.5 scenarios. Then, we evaluated the performance of these models against historical observations using the root mean square error (RMSE) to identify those suitable for China (Table S1). Subsequently, based on the selected models, we constructed a multimodel ensemble of annual precipitation, which was used to establish the historical baseline and to project future landslide susceptibility under different climate scenarios (SSP1-2.6, SSP2-4.5, and SSP5-8.5) and future periods, including the near-term future (2026–2050), mid-term future (2051–2075), and long-term future (2076–2100). To further





account for uncertainty in rainfall-induced landslides across China, we applied a 95% confidence interval derived from the T-distribution. Then, landslide susceptibility was classified into five levels using the Jenks natural breaks method. To ensure comparability of future results, we applied the classification thresholds derived from the baseline period to all subsequent assessments under different future periods and scenarios. Finally, we assessed the impact of climate change on landslide susceptibility by analysing the changes in susceptibility relative to the baseline period across the different future periods and

climate change scenarios.

## 4. Results

### 4.1 Selection of conditioning factors

First, Pearson correlation coefficients were used to select the most suitable landslide influencing factors from 12 continuous variables. The correlation coefficients among these 12 factors are shown in the correlation matrix (Fig. 3(a)). A strong positive

correlation was observed between plan curvature and curvature ($r = 0.85$, $p < 0.05$); therefore, curvature was removed to avoid multicollinearity. To further verify whether the remaining 11 continuous variables and 2 categorical variables were suitable for landslide susceptibility modelling, multicollinearity tests were conducted (Fig. 3(b-c)). All factors had TOL values greater than 0.1 and VIF values less than 10, indicating no multicollinearity among the 13 influencing factors. The final selected factors included annual precipitation, TWI, elevation, 475-year RP PGA, NDVI, plan curvature, profile curvature, slope,

distance to road, distance to fault, distance to river, landcover, and lithology.

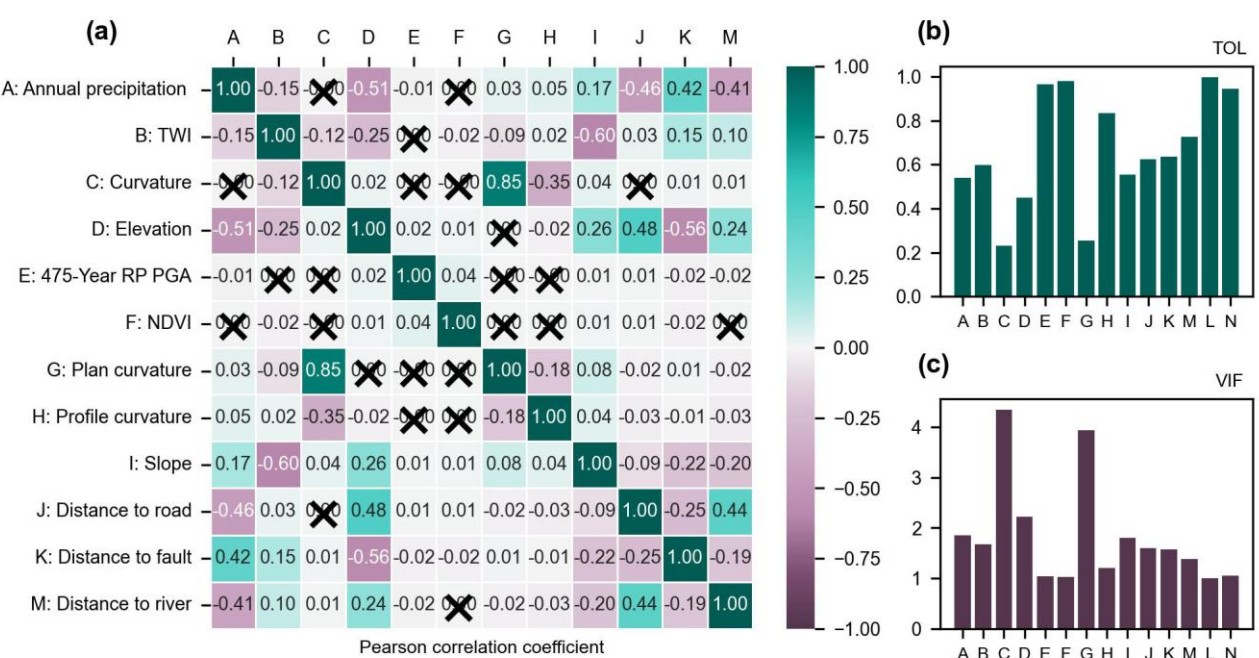





**Figure. 3:** The results of feature selection in this study: (a) Pearson correlation coefficients between landslide influencing factors ('×' indicates no significant linear relationship [p value > 0.05]) (b) TOL and VIF for different landslide influencing factors. M: Landcover, N: Lithology.

## 4.2 Model evaluation

Model performance evaluation is a crucial component in the analysis of future rainfall-induced landslide susceptibility. In this study, 30% of the sample data were utilized as a test set to assess the performance of the RF model. As illustrated in Fig. 4, model performance was visually analysed through the ROC curve and the confusion matrix. The RF model demonstrated exceptional predictive performance, with an AUC value as high as 0.97. Fig. 4(b) indicates that the number of TP and TN significantly outweighed the number of FP and FN, suggesting that the majority of landslide samples were accurately classified and reflecting the model's robust classification ability. In the test set, the model achieved an accuracy of 0.91, a precision of 0.84, a recall of 0.89, and an F1 score of 0.87, further validating its robust predictive performance. Additionally, through the analysis of Gini coefficients in the RF model, we explored the interpretability of the model. The results revealed that annual precipitation is the key factor influencing model predictions, with a contribution rate as high as 26%, significantly higher than other factors. Elevation and NDVI contributed 12% each (see Fig. S1).

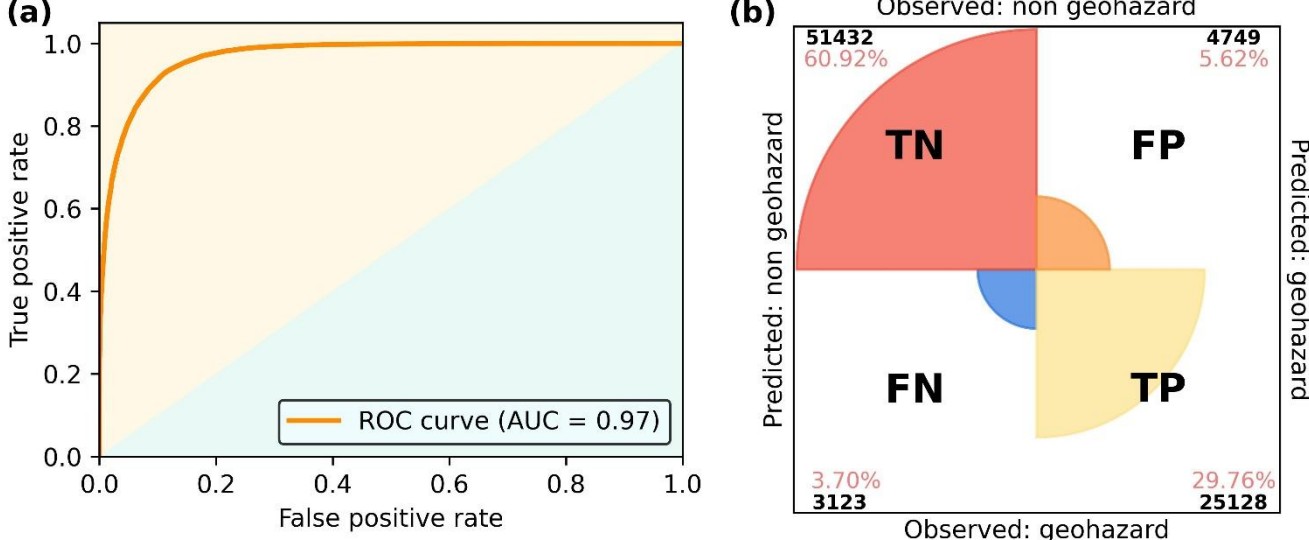

**Figure. 4:** Performance evaluation of the RF model. (a) The ROC curve demonstrates the model's classification capability, with an AUC of 0.97. (b) The confusion matrix provides a detailed breakdown of the prediction results, including the numbers of correctly and incorrectly classified samples in each category.



**4.3 Future precipitation changes in China**

Fig. 5 shows projected changes in mean annual precipitation across China and its geological environmental zones under three climate scenarios and future periods, relative to the baseline. Overall, multimodel ensemble simulations show that future mean annual precipitation across China exhibits an increasing trend. The relative increases and associated uncertainties are greater under the SSP5-8.5 than under the SSP1-2.6 and SSP2-4.5. Compared to the near-term (2026–2050) and mid-term future (2051–2075), the long-term future (2075–2100) generally shows larger increases. According to projections from the NEX-

GDDP-CMIP6 multimodel ensemble, mean annual precipitation in China is expected to increase by 59 mm, 63 mm, and 111 mm under the SSP1-2.6, SSP2-4.5, and SSP5-8.5, respectively, during the long-term future from 2076 to 2100. These correspond to increases of 10.46%, 11.18%, and 19.14% relative to the baseline.

Projected changes in mean annual precipitation exhibit substantial spatial heterogeneity across different geological environmental zones. In the NW, where the baseline mean annual precipitation is only 136 mm, the relative increase in future

mean annual precipitation remains low. During the long-term future under the SSP5-8.5, the increase is 59 mm. In geological environmental zones where baseline mean annual precipitation ranges from 400 to 900 mm, the relative increases in mean annual precipitation are generally higher, particularly in the Tibet and Yangtz. For instance, during the long-term future, under the SSP5-8.5, the largest increases are projected in these regions, reaching 170 mm and 147 mm, respectively. In the Southern region including SC and SW, where baseline mean annual precipitation exceeds 1000 mm, the absolute increase is relatively

smaller compared to areas with baseline mean annual precipitation between 400 and 1000 mm. In the geological environment division, the associated uncertainties are the highest.





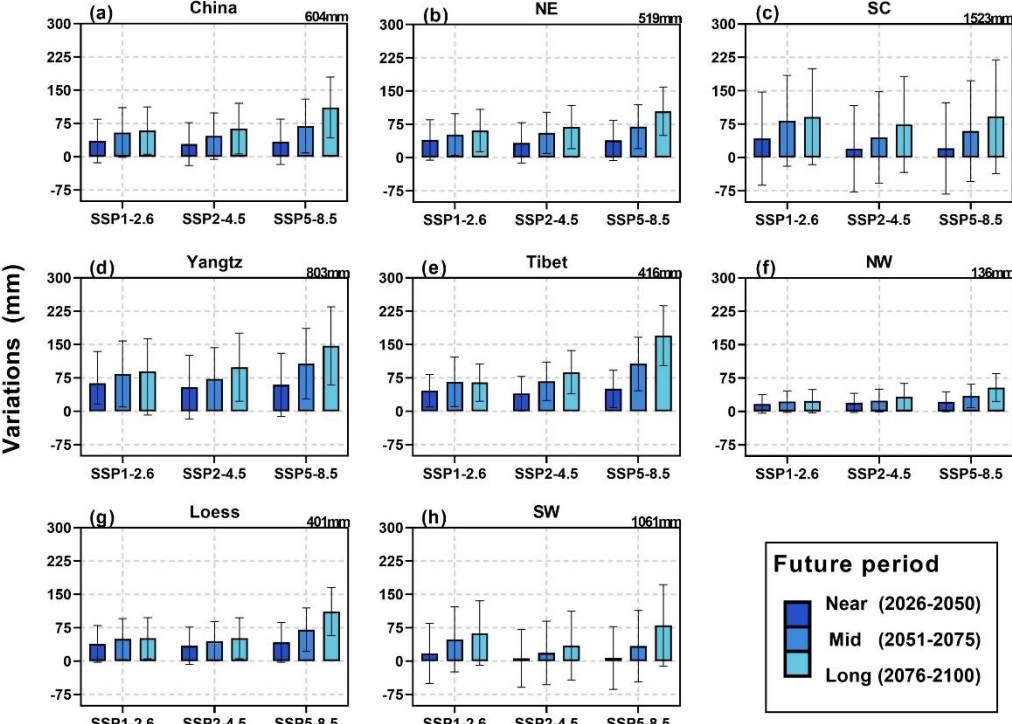

**Figure. 5:** Changes in multimodel mean annual precipitation under different future scenarios and time periods relative to the

baseline (1950–2014). Bars represent the mean change derived from the multimodel ensemble, while error bars indicate the 95% confidence interval, reflecting the uncertainty in projected mean annual precipitation. The value in the upper right corner of each panel represents the mean annual precipitation during the baseline period.

## 4.4 Future landslide susceptibility in China

Fig. 6(a) illustrates the spatial distribution of annual landslide susceptibility during the baseline period (1950–2014). The results show that approximately 43.8% of the simulated area in China exhibited very low susceptibility, primarily located in northern Tibet, southern NW, northwestern Loess, and central NE. Regions with median to very high susceptibility accounted for 21.0% of the total simulated area, mainly distributed across SC, SW, and the Loess–Taihang Mountain area. Notably, in the SC, approximately 80% of the area was exposed to median to very high landslide susceptibility.

Fig. 7 illustrates the increase in the area of median to very high landslide susceptibility zones in China compared to the baseline period across future scenarios and periods. At the national scale, the spatial extent of median to very high susceptibility zones is projected to expand under all future scenarios and periods. The expansion is more pronounced under SSP5-8.5 compared to SSP1-2.6 and SSP2-4.5. In addition, long-term future shows a more significant increase in susceptibility area than near-term




and mid-term future. In the baseline period, the proportion of median to very high susceptibility areas in China is 21%. During

the long-term future under the SSP5-8.5, it increases to 23%, corresponding to a spatial expansion of approximately $2.2 \times 10^5$ km²

At the regional scale, all geological environmental zones show an increasing trend in the extent of median to very high landslide susceptibility areas. The Loess and the SW exhibit the most significant expansion (Fig. 6 and Fig. 7). In the baseline period, the proportion of areas with median to very high susceptibility in these regions is 7.7% and 73.9%, respectively. During the

long-term future under the SSP5-8.5, these proportions increase to 18.1% and 78.2%, corresponding to area increases of $9.2 \times 10^4$ km² and $5.0 \times 10^4$ km². These expansions mainly occur in northern SW, and around the Taihang Mountains in the Loess (Fig. 6 and Fig. S2). During the long-term future under the SSP5-8.5, other regions such as Tibet, Yangtz, SC, and NE also show notable increases. According to the multimodel ensemble averages, the proportion of median to very high susceptibility areas in these regions increase by approximately $2.2 \times 10^4$ km², $1.8 \times 10^4$ km², $1.6 \times 10^4$ km², and $1.1 \times 10^4$ km², respectively,

compared to the baseline. The expansions are mainly located in southwestern Tibet and its boundary with the Loess, central and southern Yangtz, northern SC, and southern NE. The NW shows no significant change in median to very high susceptibility areas based on the ensemble mean. Most changes here involve transitions from very low to low susceptibility, primarily concentrated in the western and southeastern parts of the NW (Fig. 6).

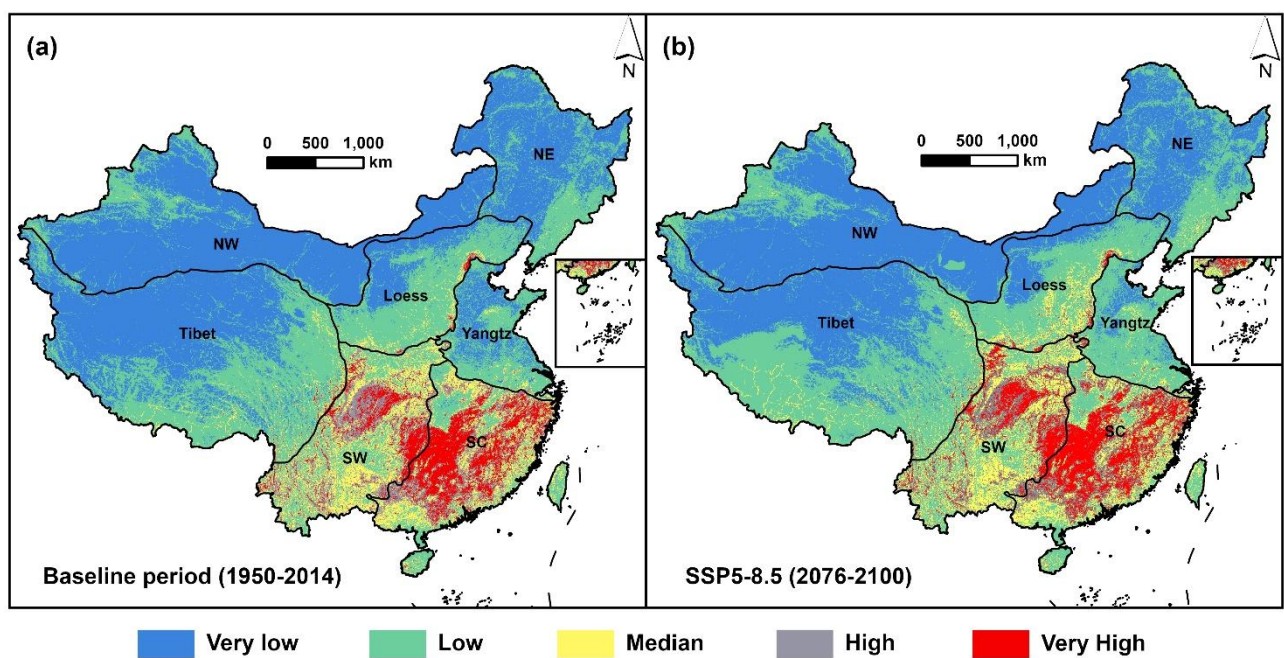


**Figure. 6:** Spatial pattern of landslide susceptibility in China during (a) the baseline period (1950–2014), and (b) the long-term future (2076–2100) under the SSP5-8.5.




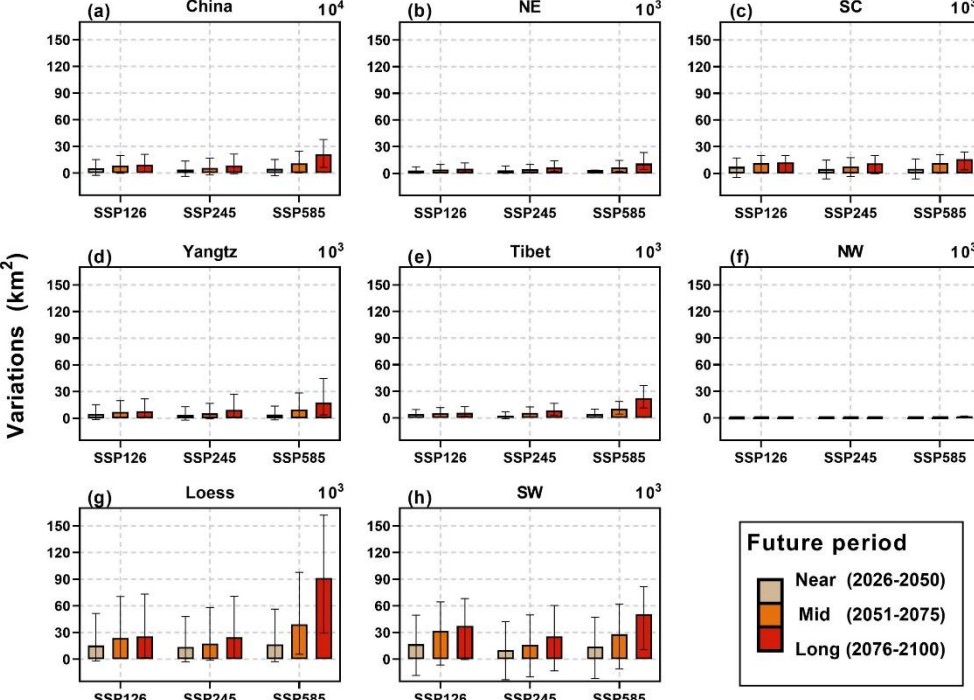

**Figure. 7** Changes in the area of regions with median to very high landslide susceptibility under different future periods and SSPs, relative to the baseline period. Bars represent the multimodel ensemble mean changes, and error bars indicate the 95% confidence interval of the ensemble mean.

## 4.5 Influence of key factors on landslide susceptibility

We further analysed the relationship between landslide susceptibility and key influencing factors in China in the far-future under the SSP5-8.5 scenario. The top six continuous variables identified by RF importance ranking were categorized, and the area proportions of each category within different landslide susceptibility classes were calculated (Fig. 8). The results show that both mean annual precipitation and NDVI are strongly positively correlated with landslide susceptibility. In areas with mean annual precipitation greater than 1200 mm, the proportions of very high, high, and median susceptibility zones are 78.34%, 59.19%, and 48.64%, respectively (Fig. 8a). Similarly, in regions where NDVI exceeds 0.5, these proportions are 76.61%, 76.94%, and 67.17%, respectively (Fig. 8c). These findings indicate that denser vegetation cover and intense rainfall significantly increase the likelihood of high landslide susceptibility. Elevation shows a strong negative correlation with landslide susceptibility. In areas with elevation between 0 and 1000 m, 92.56%, 77.73%, and 53.74% of the very high, high, and median susceptibility zones are distributed, respectively (Fig. 8b). In addition, landslide susceptibility is higher near roads,



with 76.59%, 69.70%, and 64.19% of the very high, high, and median susceptibility areas located within 10 km of a road (Fig. 8d). Landslide susceptibility decreases with increasing distance from faults within 0–200 km, but increases sharply beyond 200 km (Fig. 8f). The 475-year RP PGA generally reduces landslide susceptibility as it increases; however, when 475-year RP PGA exceeds 0.20 gal, the area proportions of very high, high, and median susceptibility zones rise again. This is because our study focuses on rainfall-triggered landslides, so the correlation with 475-year RP PGA is not strong. As a result, areas with lower 475-year RP PGA may have higher proportions of high susceptibility zones. The trend of first decreasing and then increasing somewhat reflects that some landslides may be triggered by the combined effects of rainfall and seismic activity (Fig. 8e).

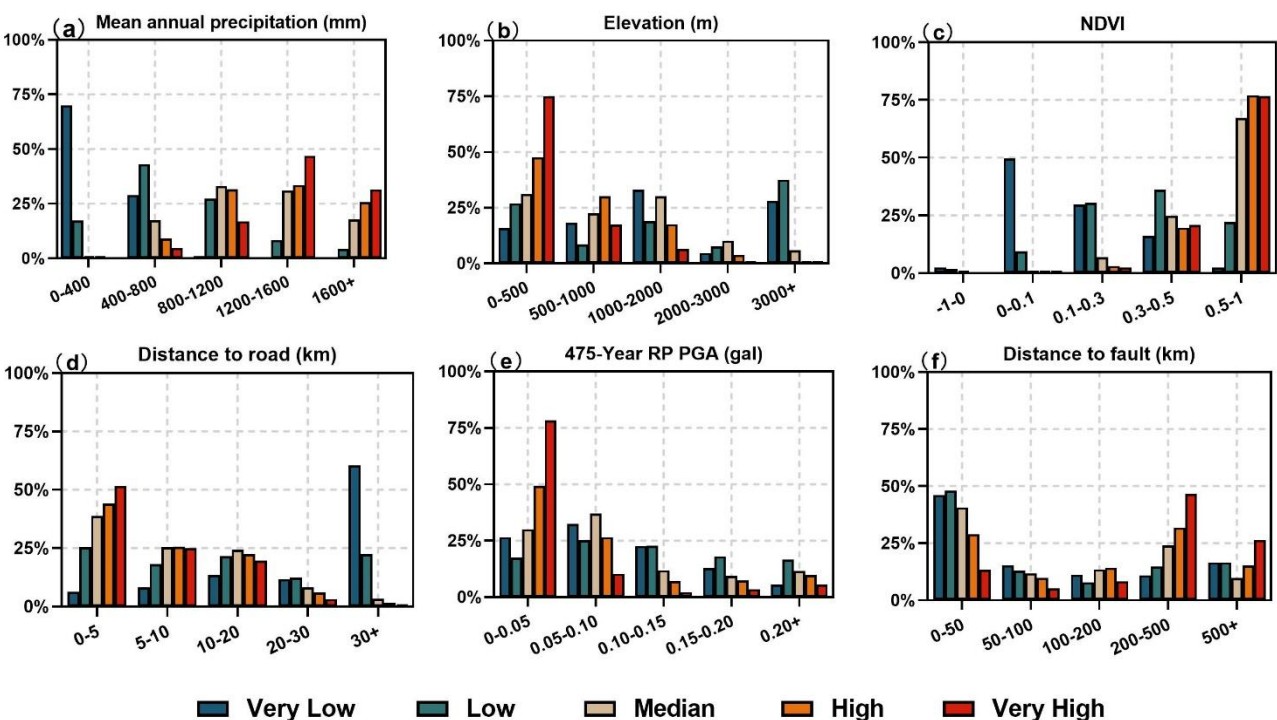

**Figure. 8:** Area proportions of landslide susceptibility classes across different categories of the top six continuous variables under the SSP5-8.5 scenario for 2076–2100: (a) Mean annual precipitation, (b) elevation, (c) NDVI, (d) distance to road, (e) 475-year RP PGA, and (f) distance to fault. Colours indicate susceptibility levels from very low to very high.

We further analysed the effects of multi-factor combinations on the spatial distribution of landslide susceptibility in China during the long-term future under the SSP5-8.5. Using the top three variables ranked by feature importance derived from the RF model (mean annual precipitation, NDVI, and elevation), we calculated the area proportion of regions with median to very high landslide susceptibility under different combinations, and visualized the results as heatmaps (Fig. 9). The results indicate



that the proportion of median to very high susceptibility areas generally increase with higher mean annual precipitation and NDVI values. Notably, when NDVI exceeds 0.5 and mean annual precipitation is 1200–1600 mm, the proportion reaches

83.89% in this range, followed by a slight decline (Fig. 9a). The analysis of elevation and mean annual precipitation shows that low-elevation areas with high mean annual precipitation have a larger proportion of median to very high susceptibility. Specifically, in areas with elevations between 500 and 1000 m and mean annual precipitation of 1200–1600 mm, the highest proportion (90.90%) is observed (Fig. 9b). These regions are river valleys and hilly terrain, where thick loose materials combined with high precipitation may significantly increase slope instability risks. In contrast, in high-altitude regions (>3000

m), even with abundant precipitation, the proportion remains below 20%, which may be attributed to lower temperatures, sparse vegetation, and frequent freeze-thaw cycles that suppress landslide occurrence and development. The combined analysis of NDVI and elevation further reveals that areas with higher NDVI and relatively low elevation are more likely to have higher proportions of median to very high susceptibility. The highest proportion (74.04%) is observed in areas where NDVI is greater than 0.5 and elevation is 500–1000 m (Fig. 9c). Overall, regions with abundant precipitation, dense vegetation, and low

elevation are expected to be the primary areas of median to very high landslide susceptibility in the future.

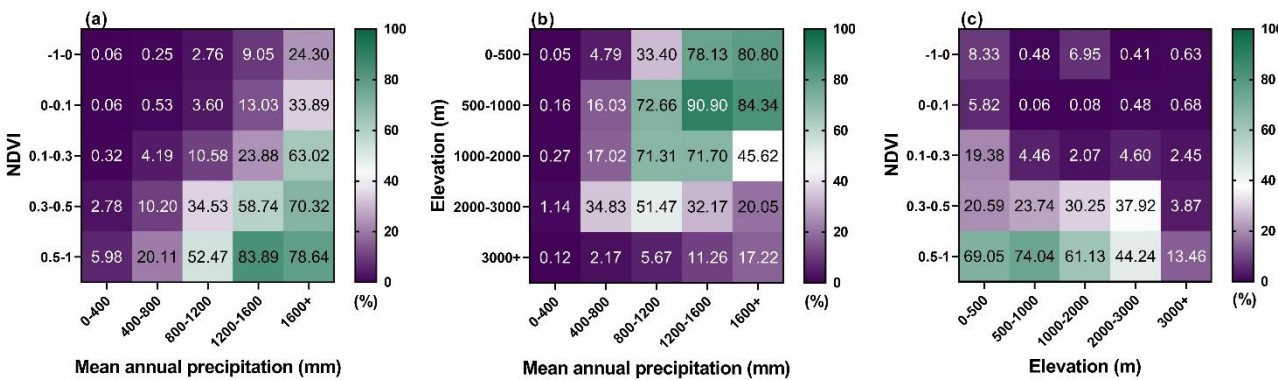

**Figure. 9:** Area proportions (%) of median to very high landslide susceptibility under the SSP5-8.5 scenario for 2076–2100, based on combinations of key continuous variables: (a) NDVI and mean annual precipitation, (b) elevation and mean annual

precipitation, and (c) NDVI and elevation. Each square in the heatmap represents the percentage of areas with median to very high susceptibility within the specific combination of mean annual precipitation and elevation ranges. Specifically, the value in each square indicates the proportion of median to very high susceptibility areas within the total area of that specific mean annual precipitation and elevation combination. The colours reflect the proportion of such areas, with specific values labelled for clarity.




## 5. Discussion

This study has investigated the future trends of rainfall-induced landslide susceptibility under various climate change scenarios. We utilized SSP1-2.6, SSP2-4.5, and SSP5-8.5 from the NEX-GDDP-CMIP6 climate dataset. A RF model, trained with a relatively complete landslide inventory, was employed for this analysis. Our projections indicate an overall increase in China's landslide susceptibility in the future compared to the baseline. This increase is particularly notable under the SSP5-8.5 scenario during the long-term future. These results are consistent with previous research. For example, Du et al. (2025) predicted increased rainfall-induced landslide susceptibility in southern Jiangxi Province using a generalised additive models (GAMs). Similarly, Guo et al., (2023) assessed a rise in shallow landslide susceptibility in Wanzhou County based on the fast shallow landslide assessment model (FSLAM). Lin et al., (2022) also found a potential increase in future landslide susceptibility and frequency across China using a generalized additive mixed effects model (GAMM). These studies collectively support our finding of a growing trend in future landslide susceptibility. Furthermore, we analysed regional variations by dividing China into seven distinct areas. Our analysis reveals specific regional changes in landslide susceptibility. For example, we found that the SW, and the Loess are key areas where landslide risk is likely to intensify. In particular, the northern SW, and around the Taihang Mountains in the Loess region are expected to witness the most significant expansion of median to very high susceptibility zones, especially under SSP5-8.5.

The accuracy of landslide inventories is a key source of uncertainty in susceptibility mapping at the national scale. It is generally assessed along three dimensions: positional accuracy, thematic accuracy, and completeness. Regarding positional accuracy, previous studies have shown that, once the landslide extent is correctly delineated, using either the scarp or a random location within the mapped extent as the locational reference does not significantly affect the overall susceptibility results (Margottini et al., 2013; Petschko et al., 2013). However, since the scarp is more readily identifiable, adopting it as a consistent locational reference can reduce positional uncertainty and thereby improve the spatial accuracy and reliability of susceptibility mapping. Therefore, the landslide points in our study are based on the scarp as the locational reference. Thematic accuracy relates to the accuracy of classification attributes, including landslide type, size among others. Small landslides, due to their limited spatial extent and ephemeral morphological expression, are often difficult to capture in national-scale inventories, leading to systematic omissions or under-reporting. To address this issue, modeling and mapping were performed for China after randomly removing 50% and 100% of the small-landslide samples, respectively (Fig. S3). Relative to the results based on the complete inventory, the exclusion of small-landslide records had a pronounced effect on susceptibility patterns: in SC, susceptibility levels decreased significantly, whereas in other regions susceptibility tended to increase overall. When only 50% of small landslides were removed, the overall susceptibility pattern remained highly consistent with that from the complete inventory, yet susceptibility was still overestimated on the whole. With respect to completeness, in SC and SW—where landslides are frequent and research attention is high—inventory records are relatively adequate. By contrast, in NW, NE, and Tibet, constrained by terrain conditions, data accessibility, and limited monitoring capacity, landslide samples are relatively scarce (Fig. S4) (Liu et al., 2013; Liu and Miao, 2018). To examine this issue, we randomly removed 50% and 75% of the



samples in NW, NE, and Tibet and then performed modeling and mapping (Fig. S5). Relative to the complete-inventory results,
these sample-poor regions exhibited an overall decrease in susceptibility as the degree of sample removal grew, whereas
regions with sufficient samples showed almost no change. Based on the above experiments, in the compilation of national-
scale landslide inventories, we suggest adopting the scarp as the locational reference, prioritizing the accuracy of thematic
classification, and subsequently supplementing and refining the records to improve the robustness and reliability of
susceptibility-mapping results.

Currently, CMIP6 GCMS are widely used to explore the impact of climate change on landslides. This study integrates the
latest NEX-GDDP-CMIP6 GCM dataset, encompassing SSP1-2.6, SSP2-4.5, and SSP5-8.5. This dataset offers globally
downscaled scenarios from CMIP6 simulations at a $0.25° \times 0.25°$ resolution. It employs bias correction and spatial
disaggregation techniques, incorporating ground observations and raw GCM data for downscaling (Thrasher et al., 2022),
thereby enhancing resolution while maintaining long-term future trends (Thrasher et al., 2012). This higher-resolution future
climate data enables a more precise assessment of precipitation-induced landslide susceptibility under climate change. First,
we selected climate models suitable for the China region by comparing the discrepancy between historical observed and
simulated precipitation using the RMSE. Following this selection, we employed a multimodel ensemble method to project
future landslide susceptibility. To quantify the associated uncertainty, we used a 95% confidence interval based on the t-
distribution. These steps are anticipated to reduce uncertainty in our future landslide susceptibility predictions. Furthermore,
this study utilizes future climate data to analyse landslide susceptibility under low, medium, and high socioeconomic
development scenarios. This analysis can be combined with future population and economic conditions to assess the potential
impact and risks of future landslides on human society (Emberson et al., 2020; Gariano and Guzzetti, 2016). Consequently,
the findings of this research provide a robust data foundation for further investigation into the potential socioeconomic risks
and impacts associated with future landslides.

This study systematically assesses the spatial distribution and future trends of rainfall-induced landslide susceptibility in China
and its geological environmental regions under climate change at a national scale. By integrating a relatively complete national-
scale landslide inventory and employing a high-resolution precipitation data from NEX-GDDP-CMIP6, uncertainties
associated with historical landslide records and future rainfall projections are reduced. The proposed method enhances the
accuracy and reliability of landslide susceptibility assessment. It provides scientific support for national-scale landslide
management and disaster mitigation strategies.

From our perspective, this study presents the following limitations: (1) This research developed an annual-scale future
landslide susceptibility model for China. However, short-duration intense rainfall is a recognized critical trigger for rainfall-
induced landslides, and analysing the seasonal variability of precipitation could provide further insights into future landslide
susceptibility changes (Lin et al., 2021). Future advancements in global navigation satellite Systems (GNSS) and aerospace
remote sensing (RS) technologies, which enable the acquisition of more precise spatiotemporal information on landslide events
and enhance the dynamic representation of landslide influencing factors, hold the potential to improve modelling reliability.
Consequently, future work could explore monthly landslide susceptibility changes in China and its geological environmental

zones under climate change, building upon improved landslide inventories and incorporating data from these advanced technologies. (2) The resolution constraints of the future climate model data introduce a limitation in predicting landslide
susceptibility across various future periods and scenarios. This may present challenges in accurately capturing localized precipitation patterns. As a result, this could lead to either underestimation or overestimation of local landslide susceptibility, thereby affecting the overall prediction uncertainty.

## 6. Conclusion

This study combined the latest NEX-GDDP-CMIP6 climate projections with a RF model to assess future changes in landslide
susceptibility across China and its geological environmental zones under different climate scenarios. The model demonstrated good performance, with an overall accuracy of 0.91, precision of 0.84, recall of 0.89, F1-score of 0.87, and an AUC of 0.97. These results indicate that the model effectively captures spatial patterns of landslide susceptibility. Among all variables, annual precipitation contributed the most to the model's output, with a relative importance of 26%. Under the selected scenarios, China's mean annual precipitation is projected to increase by 59 to 111 mm by the end of the 21st century. Compared to the
baseline period, this change is expected to result in a 4.3% to 10.6% relative increase in the area classified as having median to very high landslide susceptibility. These increases are more significant in the future long-term future and under SSP5-8.5. Spatially, several regions are projected to become more vulnerable. Notably, the SW and the Loess are key areas where landslide risk is likely to intensify. In particular, the northern SW, and around the Taihang Mountains in the Loess region are expected to witness the most significant expansion of median to very high susceptibility zones. These regions should be
prioritized in future landslide risk assessments and climate change adaptation strategies.

## Data availability

Data that support the findings of this study are available from the corresponding author upon reasonable request.

## Author contributions

J.W. designed the research, conducted the formal analysis, and wrote the original draft. H.F. provided the data and related
resources. K.L. contributed to methodology, visualization, review, and funding acquisition. Y.Y. contributed to editing the manuscript. M.W. supervised the research. B.L. was responsible for software development and validation. X.M. contributed to validation.

## Competing interests

The contact author has declared that none of the authors has any competing interests.



## Special issue statement

This manuscript is submitted to the NHESS Special Issue titled "The influence of landslide inventory quality on susceptibility and hazard map reliability" (1 July 2024 – 1 April 2026), edited by Santangelo et al.

## Financial support

This work was supported by the General Program of National Natural Science Foundation of China (grant number 42377467) and the Fundamental Research Funds for the Central Universities (grant number 2243300007).

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
