# Peer review of "Projecting changes in rainfall-induced landslide susceptibility across China under climate change"

_EGUsphere, 2025_

## Author Comment (AC2)

**Response letter for**
**"Projecting changes in rainfall-induced landslide susceptibility across China under climate change"**
**(egusphere-2025-3834)**

**Response to Reviewer1 Comments**

Reviewer #1: The proposal of the manuscript is interesting; however, it presents several points that warrant consideration:

Thank you for your thoughtful and constructive comments on our manuscript. We have carefully considered your feedback and made the necessary revisions while ensuring that the overall content and structure of the manuscript remain intact. It has been an honour to benefit from your expertise throughout this process, and we sincerely appreciate your valuable insights, which have significantly contributed to improving the quality of our work.

We have included the comments in this letter and responded to them individually. The revisions are highlighted in yellow in the revised manuscript, and the response is listed below in blue. The in-text citations in our response are marked in orange.

Sincerely,
Jinqi Wang, on behalf of all co-authors.

1. The rationale for utilizing such contrasting and extreme IPCC precipitation projection scenarios—specifically SSP1-2.6 (very low) versus SSP5-8.5 (very high)—is unclear.

**Response:**

We thank the reviewer for this helpful comment and agree that the rationale for scenario selection should be stated more clearly. In this study, we considered three SSP scenarios: SSP1-2.6 (sustainable development path), SSP2-4.5 (middle-of-the-road path), and SSP5-8.5 (high carbon emission path).

These three scenarios were selected to provide a representative range of future socioeconomic and radiative forcing conditions, thereby spanning a broad range of plausible future precipitation changes. This design enables us to examine how rainfall-induced landslide susceptibility may evolve under contrasting but plausible climate pathways and to characterize the range of possible national-scale outcomes. The corresponding analysis have been added in the revised manuscript (Lines 145–159, 507–526).

2. Similarly, the justification for employing such long-term time horizons (e.g., 2051–2075; 2076–2100) is questionable. This period is significantly distant from our current reality, limiting the applicability of meaningful interventions, and is subject to substantial uncertainty.

**Response:**

We thank the reviewer for this thoughtful comment. We acknowledge that future projections involve a

certain degree of uncertainty, particularly for later periods. However, the use of mid- and late-21st-century time horizons (e.g., 2051–2075 and 2076–2100) follows the standard time-slice framework widely adopted in IPCC/CMIP6-based climate-impact assessments, and it has been commonly used in previous studies to facilitate comparison across scenarios and to examine long-term changes in climate-related hazards, including rainfall-induced landslide susceptibility (Duan et al., 2025; Lin et al., 2022).

The objective of this study is to examine national-scale, long-term trends and spatial patterns of rainfall-induced landslide susceptibility under different climate pathways. From this perspective, mid- and long-term periods are necessary to capture the cumulative influence of sustained changes in precipitation regimes and to evaluate scenario-dependent differences in susceptibility evolution.

To help reduce and better constrain projection uncertainty, we adopted a multi-model ensemble approach and employed the higher-resolution downscaled NEX-GDDP-CMIP6 dataset (Thrasher et al., 2022), which preserves long-term climate-change signals while providing finer spatial detail. This consideration is described in Lines 507–526 of the manuscript.

3. Given China's vast territorial extent, analyzing the entire country practically precludes the use of a spatial resolution (level of detail) that is adequate for, and compatible with, the scale of most mass-movement events (i.e., landslides, which are often < 30 m) that typically occur globally.

Response:

Thank you for your valuable feedback. We fully understand your concern regarding the spatial resolution, especially when considering the vast expanse of China. Indeed, this presents a significant challenge. The primary objective of our study is to analyze the temporal trends of landslide susceptibility across China under future climate change scenarios. To achieve this, we opted for a relatively coarse spatial resolution (1 km × 1 km) in order to conduct a large-scale trend analysis across the entire country.

In terms of model evaluation, we used metrics such as AUC, accuracy, precision, recall, and F1 score to validate the model's performance. Our model achieved an AUC of 0.97, an accuracy of 91%, precision of 84%, recall of 89%, and an F1 score of 87% on the test set. These results demonstrate that, despite using a relatively coarse spatial resolution, our model is still able to effectively capture the overall trends in landslide susceptibility and exhibits strong predictive capability.

It is worth noting that many existing studies in this field have employed similar methods, using coarser spatial resolutions to analyze large-scale trend changes. These studies have successfully demonstrated the effectiveness of this approach at global or large regional scales, which provides strong support and reference for our work (Li et al., 2022; Lin et al., 2022).

We also recognize that conducting higher-resolution local studies in specific high-risk areas could provide more detailed insights and enhance the accuracy of local susceptibility assessments. Based on the current trend analysis, we anticipate that future research could further increase the spatial resolution and data precision, thereby improving the model's performance and accuracy. We have added this spatial-resolution limitation in the revised manuscript in the Limitation section (Lines 533–546).

4. The inclusion of certain 'Influencing Factors' (e.g., NDVI, Distance to River, and Distance to Road) seems inappropriate for an analysis focused on mass movements.

Response:

We thank the reviewer for their valuable comment. As shown in Response Fig. 1, the distribution patterns of landslide points significantly differ from those of non-landslide points with respect to these variables. Specifically, landslides are more frequently observed in areas closer to rivers and roads, and in regions with relatively higher NDVI values, whereas non-landslide points are more evenly distributed across these ranges. These contrasts indicate that the selected variables have a significant ability to distinguish landslide-prone from non-prone areas at the national scale.

From a mechanistic perspective, proximity to rivers may reflect the influence of fluvial erosion and hydrological factors on slope stability (Gulbet and Getahun, 2024), while proximity to roads typically represents the effects of engineering disturbances, such as slope cutting, additional loading, and altered drainage (Li et al., 2022). Higher NDVI values are generally associated with humid, vegetated hilly-mountain environments, which are more susceptible to rainfall-triggered landslides (Li and Duan, 2024). Therefore, these variables are treated as conditioning factors representing the environmental and anthropogenic influences on landslide-prone areas and are used as input variables in the model. The corresponding analysis have been added in the revised manuscript (Lines 393–412, 440–449).

[Figure]

**Response Figure. 1** | The relationship between landslide points and non-landslide points and conditioning factors. (a) Distance to Fault, (b) 475-year return period PGA, (c) Elevation, (d) Slope, (e) Distance to River, (f) NDVI, (g) Precipitation: Annual total precipitation, (h) Lithology, and (i) Landcover. Landslide

hazard frequency is defined as the "number of landslide samples/total sample size," where the total sample consists of hazard points and twice the number of non-hazard points.

We sincerely appreciate your valuable comments once again. If there are any further issues with the revised manuscript or if we have misunderstood any of your points, we would be grateful for your further guidance.

Best wishes!

**References**

Duan, Y., Ding, M., He, Y., Zheng, H., Delgado-Téllez, R., Sokratov, S., Dourado, F., Fuchs, S., 2025. Global projections of future landslide susceptibility under climate change. Geoscience Frontiers 16, 102074. https://doi.org/10.1016/j.gsf.2025.102074

Gulbet, E., Getahun, B., 2024. Landslide susceptibility mapping using frequency ratio and analytical hierarchy process method in Awabel Woreda, Ethiopia. Quaternary Science Advances 16, 100246. https://doi.org/10.1016/j.qsa.2024.100246

Li, B., Liu, K., Wang, M., He, Q., Jiang, Z., Zhu, W., Qiao, N., 2022. Global Dynamic Rainfall-Induced Landslide Susceptibility Mapping Using Machine Learning. Remote Sensing 14, 5795. https://doi.org/10.3390/rs14225795

Li, Y., Duan, W., 2024. Decoding vegetation's role in landslide susceptibility mapping: An integrated review of techniques and future directions. Biogeotechnics 2, 100056. https://doi.org/10.1016/j.bgtech.2023.100056

Lin, Q., Steger, S., Pittore, M., Zhang, J., Wang, L., Jiang, T., Wang, Y., 2022. Evaluation of potential changes in landslide susceptibility and landslide occurrence frequency in China under climate change. Science of The Total Environment 850, 158049. https://doi.org/10.1016/j.scitotenv.2022.158049

Thrasher, B., Wang, W., Michaelis, A., Melton, F., Lee, T., Nemani, R., 2022. NASA Global Daily Downscaled Projections, CMIP6. Sci Data 9, 262. https://doi.org/10.1038/s41597-022-01393-4